# Trace Elements and Omega-3 Fatty Acids of Wild and Farmed Mussels (*Mytilus galloprovincialis*) Consumed in Bulgaria: Human Health Risks

**DOI:** 10.3390/ijerph181910023

**Published:** 2021-09-23

**Authors:** Katya Peycheva, Veselina Panayotova, Rositsa Stancheva, Lubomir Makedonski, Albena Merdzhanova, Nicola Cicero, Vincenzo Parrino, Francesco Fazio

**Affiliations:** 1Department of Chemistry, Medical University of Varna, 9002 Varna, Bulgaria; peytcheva@hotmail.com (K.P.); veselina.ivanova@hotmail.com (V.P.); rositsa.stancheva@mu-varna.bg (R.S.); lubomir60@yahoo.com (L.M.); a.merdzhanova@gmail.com (A.M.); 2Department of Biomedical and Dental Sciences and Morphofunctional Imaging, University of Messina, 98168 Messina, Italy; 3Science4Life, Spin off Company, University of Messina, 98168 Messina, Italy; 4Department of Chemical, Biological, Pharmaceutical and Environmental Sciences, University of Messina, 98166 Messina, Italy; vparrino@unime.it; 5Department of Veterinary Sciences, University of Messina, 98166 Messina, Italy; ffazio@unime.it

**Keywords:** trace elements, fatty acids, Black Sea, mussel, risk–benefit, human health risk

## Abstract

The unique, closed ecosystem of the Black Sea is of significant global importance. The levels and health risk of some trace elements (As, Cd, Cr, Cu, Fe, Ni, Pb and Zn) in wild and farmed mussels (*Mytilus galloprovincialis*) collected from the Bulgarian part of the Black Sea were determined and using different approaches such as Estimated Daily Intake (EDI), Target Hazard Quotient (THQ), Hazard Index (HI), Target risk (TR), human health risk levels were assessed. The mean maximum concentrations of the elements Cd, Cr, Cu, Fe, Ni, Pb and Zn in all mussel samples were below the maximum permissible limits (MPLs) except that which exceeded the limit of 2.00 mg/kg ww. Eicosapentaenoic (EPA, 20:5n-3) and docosahexaenoic acid (DHA, 22:6n-3) were the major polyunsaturated fatty acids. The fatty acids profile studied mussels showed that the farmed mussels had higher PUFA/SFA ratios, DHA and EPA + DHA content and lower SFA, AI and TI values. The target risk (TR) values for Pb, Cr, Ni and As were calculated, evaluated and showed acceptable or negligible levels. Target hazard quotients (THQs) and hazard index (HI) from elemental intake were below 1 indicated no hazard from consumption. The benefit–risk ratio indicated that wild and cultured *M. galloprovincialis* are safe for human consumption.

## 1. Introduction

Seafood is the main dietary source of omega-3 long-chain polyunsaturated fatty acids (n-3 LC-PUFAs) for humans [1,2,3,4,5,6]. Excess fishing of wild resources had led to reducing natural stocks to critical levels. To meet the growing demand for n-3 LC-PUFA, aquaculture is the seafood supply to generate high quality products on a solid basis of environment sustainability [6,7]. The n-3 LC-PUFAs have been associated with a number of beneficial health effects, including cardiovascular, cognitive, bone, and muscle health [8]. Marine bivalves are an important resource worldwide since they do not need formulated feeds, filtering phyto- and zooplankton, bacteria and organic detritus present naturally in the water column [9]. The lipid composition of any marine organism is an important indication of meat quality and permits assessing its nutritional value. 

The Mediterranean mussel (*Mytilus galloprovincialis*) is a commercially important species and is widely distributed in the Adriatic Sea, the Mediterranean, and the Atlantic coast, as well as the region of the Black Sea. In the Black Sea, *M. galloprovincialis* is harvested via traditional fishing and extensively cultured in coastal waters. 

Being filter-feeding organisms, mussels are used as bioindicators and assest the status of chemical contamination in many countries [10,11,12]. *Mytilus galloprovincialis* are sentinels, which are considered one of the most suitable indicators of pollution since they have a sedentary nature, sampling easily, capacity for accumulating contaminants, wide geographical distribution, and sensitivity to chemical pollution [9]. Under various conditions (such as stress or elements exposure), this bivalve species show sensitive and biological responses.

Trace elements are generated in the surrounding environment through various anthropogenic pressures and are easily transported through rivers and by air, reaching the marine waters where they affect the coastal areas [10,13,14]. Then they may alter the local population and aquatic ecosystems. For seafood safety, it is indeed needed to understand the elemental accumulation processes in marine and coastal ecosystems as well as the potential health risk to humans through the consumption of shellfish.

Bearing in mind the potential risk for consumers to get a certain dose of trace elements, and the health benefits of seafood on the other hand, it is of great interest to assess the benefit–risk ratio of consumption of a portion of a given species and thus to provide consumers with relevant recommendations [15,16,17,18,19,20,21,22,23,24,25,26,27,28,29]. The elements may be classified as essential (copper, selenium, zinc, manganese, iron), probably essential (cobalt and vanadium), and toxic (cadmium, arsenic, mercury, lead, and nickel). At higher concentrations, essential elements may also produce a toxic effect. [14]

This paper studies (1) the toxic (As, Cd, Ni, and Pb) and essential element (Cu, Cr, Fe, Mn, and Zn) concentrations and fatty acid profiles of farmed and cultured *M. galloprovincialis* from the Bulgarian part of the Black Sea; (2) the impact of *M. galloprovincialis* on human health assessed by the most commonly used risk indexes (such as Target Hazard Quotient (THQ), Hazard Index (HI) and Target risk (TR)) using the maximum concentration of element that could be reached via consumption of mussels and (3) the benefit–risk ratio for human health based on trace elements and n-3 LC-PUFAs contents in mussels.

## 2. Materials and Methods

### 2.1. Chemicals

All reagents used were of analytical reagent grade. Analytical-reagent grade water was obtained by Milli-Q water system (Merck KGaA, Darmstadt, Germany) and was used for all dilution and preparation of reagents and standard solution. All laboratory glasses were soaked in dilute 2M HNO_3_, rinsed with Milli-Q water before use. 65% (*w/v*) HNO_3_ was obtained from Merck KGaA (Darmstand, Germany) and 30% (*w/v*) H_2_O_2_ (Fisher Scientific, Pittsburgh, PA, USA) was also used. The standard calibration solutions used for calibration of the trace elements were prepared by diluting an ICP multi-elements standard solution IV (1000 mg/L in 2% HNO_3_) obtained from Merck KGaA (Darmstadt, Germany).

All standards (Supelco 37 Component FAME Mix, Supelco PUFA №3 from Menhaden oil) were purchased from Merck KGaA (Darmstadt, Germany). All solvents used (chloroform, methanol and n-hexane) were of HPLC grade and purchased from Merck KGaA (Darmstadt, Germany).

### 2.2. Sample Collection and Preparation

The samples of farmed and cultured mussels were collected from four sites (stations) along the Bulgarian Black Sea coastline (Figure 1) in 2020 out of the spawning season. The stations were selected based on the availability of mussels, natural environment, and typical ecological conditions along the Bulgarian coastal regions of the Black Sea. The farmed mussels were purchased from two of the major mussel farms along the Bulgarian coast, while the wild mussels were obtained from local fishermen in Varna and Kranevo (Bulgaria).

Approximately 2 kg of mussels from each station with comparable shell length were collected, placed into plastic bags with and brought to the laboratory in ice boxes. Around one hundred animals were taken randomly for determination of sample mean, and only those of mean length 46.65 ± 4.81 mm were further processed. Around ten batches of samples containing 10–12 mussels of one size group were chosen, dissected with a Teflon knife to shells and soft tissues. After dissection, the soft tissues were rinsed with Milli-Q water to remove different particles and stored in polyethylene bags at −20 °C until analysis.

### 2.3. Trace Element Analysis

#### 2.3.1. Chemical Analysis 

Each tissue sample (around 1 g wet weight) was weighed, placed in Teflon digestion vessels. Acid wet digestions using 8 cm^3^ HNO_3_ (65% *w/v*) and 2 cm^3^ H_2_O_2_ (30% *w/v*) were performed using a microwave closed-vessel digestion system MARS 6 (CEM Corporation, Matthews, NC, USA) subject to 3 stage program, the maximum temperature of 210 °C, pressure 800 psi and maximum power 1050 W. The digested mollusk samples were cooled to 30 °C, diluted to 25 mL with Milli-Q water, and stored in polyethylene bottles until analysis.

The concentrations of As, Cd, Cr, Cu, Fe, Mn, Ni, Pb, and Zn in the samples were determined using ICP-OES Spectrometer (Optima 8000, Perkin Elmer, Waltham, MA, USA) with plasma gas flow—10 L/min, nebulizer gas flow 0.2 L/min, auxiliary gas flow—0.7 L/min and axial plasma view. The accuracy of the determination of trace elements in mussels was tested using ERM-CE 278k (mussel tissues from the European Commission, Joint Research Center, Geel, Belgium) certified reference material. The CRM was digested and analyzed in the same way as the analytical samples. The majority of elements were in the range of 10 to 19% of deviation between certified and determined values.

#### 2.3.2. Analysis of Data

The Microsoft Office Excel 2010 software with significance at *p* < 0.05 was used for the descriptive statistics and one-way analysis of variance (ANOVA).

#### 2.3.3. Human Health Risk Assessment 

The human health risk assessment through consumption of mussels was evaluated on the basis of estimated daily intake (EDI), target hazard quotient (THQ), and hazard index (HI). For the calculation of EDI, the following was used:(1)$$\mathrm{EDI}=\frac{(M_{c}\times {\mathrm{CR}}_{d\;)}}{B_{w}}\;$$where EDI is estimated daily intake (mg kg^−1^ bw day^−1^), M_c_—the average concentration of trace element (mg/kg), CR_d_—average daily consumption rate (0.8 g/capita/day in 2013 [7] (kg/person), Bw—body weight (kg). In this study, a nominal 70 kg was used as the average body weight for adults (group 18–25 age) [24].

US EPA Region III Risk-Based Concentration Table [25] was used to calculate the THQ developed by USEPA and expressed as the ratio of exposed trace element concentration to the reference dose concentration and gives information about the long-term non-carcinogenic exposure probabilities.
(2)$$\mathrm{THQ}=\frac{\mathrm{EF}\times \mathrm{ED}\times \mathrm{CR}\times C\times 10^{-3}}{\mathrm{RfD}\times B_{w}\times {\mathrm{AT}}_{n}}$$where EF is the exposure frequency (350 days/year); CR is the mean daily consumption of mussels (g/day) (0.8 g/person/day) [7]; ED is the exposure duration (30 years); C is the metal concentration in mussel species (mg/kg ww), RfD is the reference oral dose of individual element (3 × 10^−4^ mg/kg/day for As; 1 × 10^−3^ mg/kg/day for Cd, 3 × 10^−3^ mg/kg/day for Cr; 4 × 10^−2^ mg/kg/day for Cu; 7 × 10^−1^ mg/kg/day for Fe; 2 × 10^−2^ mg/kg/day for Ni; 2 × 10^−3^ mg/kg/day for Pb; and 3 × 10^−1^ mg/kg/day for Zn) [25], B_W_ is the adult body weight (average 70 kg) and AT_n_ is the average time for non-carcinogens (ED*365) [25]. 

Hazard index (HI) is an index that estimates the combined effect of contaminants, and it was calculated using the formula [25]:(3)$$\mathrm{HI}=\overset{n}{\underset{i=1}{\sum }}{\mathrm{THQ}}_{i}$$

Target cancer risk (TR) is used to evaluate the carcinogenic risks to an individual to develop cancer over a lifetime exposure to potential carcinogens [26] and it is calculated as:(4)$$\mathrm{TR}=\frac{C\times \mathrm{CR}\times 10^{-3}\times {\mathrm{CPS}}_{o}\times \mathrm{EF}\times \mathrm{ED}}{B_{w}\times {\mathrm{AT}}_{c}}$$where C is the metal concentration in mussel species (mg/kg ww), CR is the average daily consumption of mussels (g/day) (0.8 g/person/day) [7]; CPS_o_ is the carcinogenic potency slope oral (risk per mg/kg/day) with values 1.7 mg/kg bw/day for Ni; 0.5 mg/kg bw/day for Cr and 0.009 mg/kg bw/day for Pb; AT_c_ is the averaging time carcinogens (25,550 days) [24,26,27,28].

### 2.4. Fatty Acid Analysis 

Total lipids (TL) were extracted from *M. galloprovincialis* using the Bligh and Dyer procedure [30]. Three replicates (20 mussels each) were homogenized with a laboratory blender (Isolab Laborgeräte GmbH Co., Eschau, Germany). Three grams of tissue homogenates were then extracted sequentially with chloroform/methanol (1:2 *v/v*), chloroform/methanol (1:1 *v/v*) and chloroform. Constant mixing for 30 min was applied after each extraction cycle. Then, NaCl solution in H_2_O (0.9% *w/v*) was added and phase separation was achieved after centrifugation (3500 g, 15 min). The bottom organic layer was collected with a Pasteur pipette, filtered through Na_2_SO_4_ and the solvent was evaporated to dryness by rotary-evaporator at 40 °C. The dry residues were weighed and the amounts of total lipids were determined gravimetrically. 

The fatty acids (FAs) of *M. galloprovincialis* were determined as fatty acid methyl esters (FAME) after direct transmethylation with 2% sulphuric acid in methanol [31]. FAMEs were analyzed by gas chromatography using a Thermo Fisher Scientific FOCUS chromatograph (Waltham, MA, USA) equipped with a TRACE TR-5MS (Waltham, MA, USA) capillary column (30 m × 0.25 mm × 0.25 µm) and a PolarisQ (Waltham, MA, USA) ion trap mass spectrometer. The oven temperature was programmed from an initial oven temperature of 40 °C for 4 min, followed by a rate of 20 °C/min from 40 °C to 150 °C and raised from 150 °C to 235 °C at a rate of 5 °C/min, and then from 235 °C to 280 °C a rate of 10 °C/min for 5 min. The carrier gas used was helium with a flow rate of 1 mL/min. Fatty acid identification was performed by comparing their respective retention time and mass spectrum with mass spectra of the commercial 37 Component FAME Mix standards and PUFA №3 from Menhaden oil under the same conditions of FAMEs. Individual FA was expressed both as a percentage (%) of the total amount of fatty acid. The results for EPA and DHA were calculated as mg/100 g ww using the corresponding conversion factors (XFA) for mollusks, proposed by Weihrauch et al. [32].

### 2.5. Hazard Quotient for Benefit–Risk Ratio 

Gladyshev et al. [22] proposed a formula for the benefit–risk ratio of the consumption of marine organisms based on the content of LC-PUFA and toxic/essential elements. It is estimated through a calculation using the following ratio:(5)$${\mathrm{HQ}}_{\mathrm{EFA}=}\frac{R_{\mathrm{EFA}}\times C_{\mathrm{element}}}{C\times \mathrm{RfD}\times B_{w}}$$where R_EFA_ is the recommended daily dose of essential fatty acids (EFA) for a human person (mg/day), C_element_ is the concentration of the given element (mg/kg), C is the content of EFA (EPA + DHA) in mussel (mg/g), RfD is the reference dose (mg/kg/d) and B_w_ is the average adult body weight (70 kg). A value of HQ_EFA_ less than 1 means the health benefit from bivalve consumption, and HQ_EFA_ more than 1 means the risk [22]. For the calculation of this equation, R_EFA_ = 500 mg/day [33] was used, and the values for RfD were given in Anishchenko et al. [21]. 

## 3. Results and Discussion

### 3.1. The Levels of Trace Elements in Sampled Species

The mean concentrations (mg/kg ww) of essential trace elements and toxic trace elements in muscle tissues of wild and farmed *M. galloprovincialis* are given in Table 1 and Table 2.

Fe was the most abundant of all elements, varying between 119.94 ± 3.46 mg/kg ww for *M. galloprovincialis* from VPS and 188.24 ± 5.51 mg/kg ww from DFS. The farmed mussel species showed a higher value of iron compared to wild mussels. According to Jović and Stanković [36], Fe ranged between 82.05 mg/kg dw and 450.1 mg/kg dw in the soft tissues of wild and farmed *M. galloprovincialis* collected from the Boka Kotorska Bay, Adriatic Sea and between 260.0–458.6 mg/kg dw in the same species from the northern part of Adriatic Sea, Venice, Italy [37]. Concerning Cr, the same authors [37] reported values between 2.0 and 2.4 mg/kg dw, while Liu et al. [38]—0.25–0.48 μg/g, wet weight in benthic bivalves from Laizhou Bay, China. The concentration level for this element in the studies species was below the range stated by several authors [39,40,41,42].

The concentration of Zn in tissues of mussels ranged between 17.58 ± 1.30 mg/kg ww and 29.87 ± 3.37 mg/kg ww. Peroševic et al. [43] reported Zn concentration between 14.8 mg/kg ww and 24.5 mg/kg ww in *M. galloprovincialis* at three locations in Boka Kotorska in four different seasons in 2005. The FAO/WHO [44] set a limit to daily human intake for Zn 30 mg/kg. The established maximum level for zinc in Bulgarian legislation above which bivalves’ consumption is not permitted is 200 mg/kg [45].

When in excess, Mn can have adverse neurological effects even though its essential role for humans and the data obtained from the current study are within the range stated in the literature [46,47,48,49,50].

The level of copper concentration varies from 2.01 ± 0.28 up to 2.86 ± 0.32 mg/kg ww as the wild species have higher amount of it compared to farmed mussels. Őzden et al. [41] measured the copper content in different seasons in *M. galloprovincialis* between 0.839 ± 0.049 mg/kg ww and 3.116 ± 0.052 mg/kg ww, a relatively high amount of Cu (719.48 μg/g dw) were established in a sample of mollusks from the coast of Xiangshan Bay, China [38]. 

Table 2 illustrates a high concentration of As, which was observed in the farmed sample from one of the mussel farms (2.24 mg/kg ww) and exceeded the maximum permitted level of 2.0 mg/kg ww for mussels according to Bulgarian Food Codex [44] but within the range of 1 up to 5 μg/g ww according to USFDA [51]. The data in the literature stated values between 1.942 and 7.109 μg/g ww for mollusks from the Xiangshan Bay, China [38], 0.070–1.183 mg/kg for *M. galloprovincialis* from the Marmara Sea, Turkey [41] and between 1.6 and 4.6 μg/g ww in wild mussels along the South African coastline [52]. 

Cd concentration in the literature ranged between 0.292 ± 0.008 mg/kg ww and 0.970 ± 0.043 mg/kg ww in *M. galloprovincialis* from the Marmara Sea, Turkey [41], up to 0.201 ± 0.025 mg/kg ww for the same bivalve species from the Sicilian coast, Italy [53], and between 0.09 ± 0.005 and 0.91 ± 0.02 mg/kg in *D. trunculus*, *C. gallina* and *M. galloprovincialis* in our previous study of the Black Sea, Bulgaria [11]. The current values do not exceed the limit of 1.0 mg/kg set by the European Commission [35]. 

There is no maximum permitted limit set by the EU concerning bivalve mollusks. Nickel concentration in *M. galloprovincialis* sampled from the Boka Kotorska Bay, Montenegro varies between 0.27 mg/kg ww to 1.14 mg/kg ww [43], between 0.06 and 1.69 μg/g ww for wild mussels along the South African coastline [52] and between 1.35 and 7.05 mg/kg dw in the soft tissues of wild and farmed *M. galloprovincialis* collected from the Boka Kotorska Bay [36]. The results from this study are within the range stated in the literature.

The European Commission [35] sets maximum levels of 1.5 mg/kg Pb in bivalve mollusks, which is below the recorded values for the species subject to this study. In another study of the *M. galloprovincialis* in the Bulgarian section of the Black Sea, Zhelyazkov et al. [24] found Pb levels ranging from 0.157 mg/kg up to 0.414 mg/kg ww while Peycheva et al. [11]—0.13 ± 0.03 mg/kg. 

### 3.2. Lipid Composition 

Results for total lipid (TL) content and fatty acids composition of wild and farmed mussels are presented in Table 3. 

Levels of TL in *M. galloprovincialis* ranged from 1.31 g/100g ww (farmed mussel from DFS) to 2.35 g/100g ww (farmed mussel from BSS). According to the lipid content, wild and farmed *M. galloprovincialis* from the Black Sea can be classified as lean species (<2% TL) and low-fat species (2–4% TL) [4]. Values varied significantly among sampling stations, but the lipid content of wild mussels did not differ from farmed specimens. Similar values for total lipid content of wild [4,54,55,56] and farmed *M. galloprovincialis* [3,11,57,58] were reported. 

The saturated fatty acids (SFA) accounted for 37% (KS) and 40% (VPS) in wild mussels and significantly lower content (23% and 28%) for farmed mussels. Saturated fatty acid predominated only in wild mussels‘ lipids from VPS station (SFA > MUFA > PUFA). VPS station is one of the hot spots along the Black Sea coastal zone. Its ecological state is strongly influenced by diffused sources of pollution—shipping, chemical industry, human activities. Some authors suggest that higher levels of SFA may reflect increased bacterial loads or organic detritus in mussels’ diet [59]. However, a previous study reported a predominance of SFA in *M. galloprovincialis* collected from ecologically clean regions in the Black Sea [55]. 

The mussels from the other three stations show a different distribution of FA groups: PUFA > SFA > MUFA with a remarkable predominance of polyunsaturated fatty acids, similar to the patterns reported for TL of farmed and wild *M. galloprovincialis* from the Black Sea [55,60]. 

Major fatty acid levels in marine organisms generally depend on a number of biotic and abiotic factors such as water temperature and salinity, light intensity, an abundance of food, etc. [2,61,62]. The major fatty acids in the SFA group were palmitic acid C16:0 (from 16.28 to 27.28%), followed by myristic (C14:0) and stearic acid (C18:0). 

Saturated fatty acids predominated over the monounsaturated fatty acids for all sampling stations. Significant differences (*p* ≤ 0.05) were found between mussels from different sites. In farmed mussels MUFA contents accounted for 7.60% (DFS) and 23.21% (BSS), while in wild mussels—11.89% (KS) and 31.73% (VPS). Samples from VPS and BSS displayed significantly higher MUFA contents compared to the other two stations. The major fatty acids in this group were palmitoleic acid (C16:1n-7) and oleic acid (C18:1n-9). 

The samples from the three stations (BSS, KS and DFS) showed that the majority of their lipids consisted of polyunsaturated fatty acids (PUFA). In farmed mussels, PUFA contents accounted for 47.18% (BSS) and 69.71% (DFS), while in wild mussels—27.75% (VPS) and 51.04% (KS). Our results are consistent with previously published data for *M. galloprovincialis* [3,11,55,57,58,59,63,64]. The predominant fatty acids in the PUFA group are eicosapentaenoic (EPA) (20:5n-3) and docosahexaenoic (DHA) (22:6n-3) acids. Their sum accounted for 71.5% to 84.9% of total PUFA. Marine bivalves have limited ability to synthesize PUFAs de novo [6,64] and acquire them from the food available in the region [56,65]. In our study, we found that EPA and DHA contents significantly varied among the sampling stations, but DHA prevailed in mussels from VPS, BSS, and DFA. Wild and farmed mussels from KS and DFS stations contained significantly higher amounts of EPA, compared to the other two sampling sites. Mussels from DFS were characterized by higher EPA content than DHA. 

Farmed mussels showed significantly higher contents of DHA (26.85% and 30.22% in mussels from DFS and BSS stations, respectively) compared to farmed mussels (19% for both VPS and KS stations). Since mussels cannot synthesize LC-PUFAs, DHA needs to be obtained from the diet. The higher concentrations of DHA may reflect the prevalence of zooplankton and dinoflagellates in the mussels’ diets [66], but also different gametogenesis and reproduction cycles of the bivalves from different locations [67].

The most distinguishing attribute of the nutritional value of seafood is the predominance of essential long-chain PUFAs. The health benefits related to bivalve consumption can be assessed using lipid quality indices, such as PUFA/SFA ratio, n-6/n-3 ratio, AI, TI and absolute amounts of EPA + DHA (in mg/100 g edible portion). The nutritional profiles of wild and farmed *M. galloprovincialis* lipids are presented in Table 4.

One important indicator for the lipid quality of seafood is the PUFA/SFA ratio reflecting both the effects of PUFAs and SFAs, therefore a well-balanced fatty acid composition. High intake of SFA (and lower PUFA/SFA ratio, respectively) suggests an increased risk for cardiovascular health. In our study, PUFA/SFA ratios vary from 0.69 to 3.08 and are within the recommended range of PUFA/SFA ratio 0.45–4.00 [4,55,56]. An important indicator for determining quality of lipids in the n-6/n-3 ratio [68]. High n-6/n-3 ratios in (>4) dietary lipids are considered undesirable in the human diet because of their potential to promote cardiovascular diseases. In this study, n-6/n-3 ratios ranged from 0.12 to 0.28 for wild and farmed mussels. 

The health-benefit potential of mussel lipids was assessed by two other indices: the atherogenicity index (AI) and the thrombogenicity index (TI) [69]. Higher AI and TI values may stimulate platelet aggregation and thrombus formation. Thus, lower values are considered beneficial to human health. In our study, the AI values of *M. galloprovincialis* ranged from 0.27 to 0.82. The low n-6/n-3 ratio resulting from higher n-3 content in all *M. galloprovincialis* samples in this study provided lower AI and TI. The AI values were significantly lower in farmed mussels (0.35 on average) than wild (0.72 on average); as well as IT, with average values of 0.14 in farmed mussels and 0.22 in those wild. Significantly higher AI and TI values were reported for *M. galloprovincialis* from the Mediterranean Sea [3] and the Ionian Sea [4]. 

According to the dietary guidelines [33,70], a dose of 250–500 mg (one to two servings per week) of the long-chain omega-3 fatty acids EPA + DHA is recommended for human health. In our study, the lower amounts of EPA + DHA (309.4 and 428.8 mg/100g EP) were found for wild mussels, while farmed mussels contained higher values of these n-3 LCPUFAs (556.5 and 655.6 mg/100g EP). Thus, in order to obtain the recommended intake of EPA + DHA, consumers need a smaller portion of farmed mussels (from 38.1 to 44.9 g) compared to wild mussels (of about 58.3 to 80.8 g).

### 3.3. The Assessment of Potential Human Health Risk Associated with Wild and Farmed Mussel Consumption

The target hazard quotient and hazard index are given in Table 5. THQ is a convenient coefficient for the evaluation of the risk associated with the intake of contaminated mussels. The values below 1 (THQ < 1) are associated with lower levels of exposure, which is not likely to cause any harmful effects for human health during a lifetime in the population [71].

In our study, the THQs values were below 1 for all the elements in all mussel species. Nekhoroshkov et al. [52] reported mean THQ values for wild South African *M. galloprovincialis* in the range of 0.3–0.7 for the elements Cr, Co, Zn and I contributed maximum in local HI. Jović and Stanković [36] reported THQ values for *M. galloprovincialis* from different Adriatic countries as follows Cu (0.015–0.036), Zn (0.021–0.125), Ni (0.005–0.025), Cd (0.024–0.202), Pb (0.227–0.667), Hg (0.048–0.144). THQ values for all tested elements in all samples from mussel samples indicate that no health risk is present according to regulation or literature data. The calculated HI values for adults are below the safety level of 1, indicating that toxic and essential elements contained in mollusks posed no risk for the consumers. 

The estimated TR values for Cr, As, Pb and Ni are also presented in Table 5. Cd, Cu, Fe, Zn, and Mn do not cause any carcinogenic effects, so the TR value were calculated only for the intake of Cr, As, Pb and Ni. According to the US EPA methods, the cancer risk range from 10^−4^ to 10^−6^ is considered acceptable, >10^−4^ is considered unacceptable, lower than 10^−6^ is considered to be negligible [25,26]. The results of this research indicated that the carcinogenic risk for Ni, As, Cr, Ni and Pb were acceptable or lower than the negligible level.

### 3.4. Benefit–Risk Ratio Based on the Content of LC-PUFA and Toxic/Essential Elements

The values of benefit–risk hazard quotients, HQ_EFA_, are given in Table 6. According to Gladyshev et al. [22] the quantification of benefit–risk ratio could be a more precise way of estimation of nutritive value, rather than using threshold limit values alone. The content of trace elements in the studied wild and farmed *M. galloprovincialis* did not decrease their nutritional value considering the benefit–risk ratio. Farmed and wild mussels were characterized by low HQ_EFA_ values (<1), thus posing no risk for consumers, assuring the recommended intake of EPA + DHA (500 mg per day). The benefits from EPA + DHA intake via consumption of farmed and wild *M. galloprovincialis* from the Black Sea overweigh the risk. 

## 4. Conclusions

This study aimed to assess if farmed *Mytilus galloprovincialis* could compete with the wild mussels in terms of nutritional value and human health risk. 

The concentration of essential (Cu, Cr, Fe, Zn) and toxic (As, Cd, Ni, Pb) elements were below the limits for *M. galloprovincialis* stated by European Commission [35] except the value for As for farmed DSF. EDI values for both farmed and wild *M. galloprovincialis* species were lower than published RfD values. 

The lipid and fatty acids profile of farmed and wild mussel products showed significant differences. The comparison between the studied mussels showed that the farmed mussels had a higher PUFA/SFA ratio, DHA and EPA + DHA, and accordingly, lower SFA, AI and TI values than their wild counterparts. Thus, we can conclude that commonly consumed cultured mussels are products with comparatively better nutritive value, especially in hypocholesterolemic and cardio-protective diets.

As all THQ, HI and TR values were below 1, which leads to the conclusion that consumption of farmed and wild *M. galloprovincialis* collected from Bulgarian Black Sea did not pose any risk for consumer health concerning the analyzed trace elements.

Using the HQ_EFA_ index for benefit–risk ratio of essential fatty acids vs. toxic/essential elements concentration, it has been concluded that in general, the consumption of farmed and wild Black Sea *M. galloprovincialis* proposes no risk for consumers. The relatively high levels of n-3 LCPUFA shellfish exceed the risk due to the content of trace elements. 

According to the estimated hazardous quotients and indexes, the levels of the analyzed wild and farmed mussels collected in the studied region can be considered relatively safe and suitable for consumption.

Future studies on *M. galloprovincialis* trace elements, EPA and DHA contents need to be periodically carried out in order to provide up-to-date information for benefit–risk assessment of the only representative of the Bulgarian marine aquaculture. 

## Figures and Tables

**Figure 1 ijerph-18-10023-f001:**
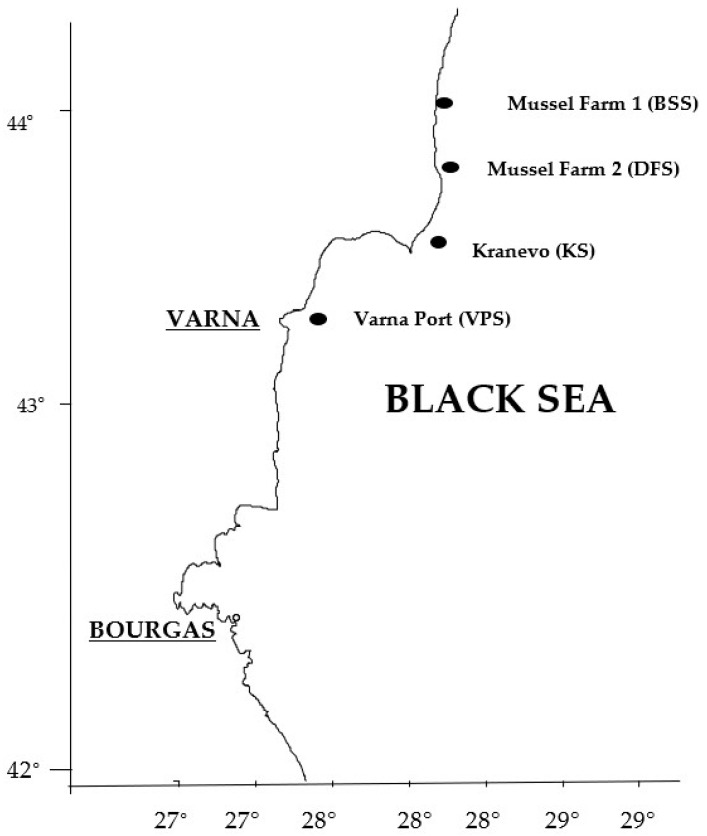
The map of sampling locations.

**Table 1 ijerph-18-10023-t001:** Essential trace element content (mean ± standard deviation) and nutritional contribution of wild and farmed *M. galloprovincialis* from the Black Sea (Bulgaria).

		VPS (Wild)			BSS (Farmed)			KS (Wild)			DFS (Farmed)		
	RDA ^1^ (mg/day)	mg/kg	EDI	% DRI	mg/kg	EDI	% DRI	mg/kg	EDI	% DRI	mg/kg	EDI	% DRI
Cr	25−35	0.15 ± 0.01	0.002	0.01	0.38 ± 0.13	0.004	0.02	0.10 ± 0.01	0.001	0.01	0.35 ± 0.08	0.004	0.02
Cu	0.90	2.86 ± 0.32	0.033	3.63	2.25 ± 0.62	0.026	2.86	2.81 ± 0.32	0.032	3.57	2.01 ± 0.28	0.023	2.56
Fe	8−18	119.94 ± 3.46	1.371	17.13	162.48 ± 31.65	1.857	23.21	121.50 ± 11.15	1.389	17.36	188.24 ± 5.51	2.199	27.49
Mn	1.8−2.3 *	2.73 ± 0.15	0.031	1.73	3.24 ± 0.54	0.037	2.047	1.56 ± 0.16	0.018	0.99	3.22 ± 0.88	0.037	2.05
Zn	8−11	17.58 ± 1.30	0.201	2.51	19.02 ± 1.53	0.217	2.72	21.45 ± 4.22	0.245	3.06	29.87 ± 3.37	0.341	4.27

^1^ RDA—Recommended Daily Allowances [34]; EDI—estimated daily intake; % DRI—calculated percent from the daily recommended intake; * AI—Adequate intake.

**Table 2 ijerph-18-10023-t002:** Toxic trace elements composition (mean ± standard deviation in mg/kg) of wild and farmed *M. galloprovincialis* from the Black Sea (Bulgaria).

	VPS (Wild)	BSS (Farmed)	KS (Wild)	DFS (Farmed)	Regulations
As	0.73 ± 0.09	1.57 ± 0.13	0.62 ± 0.04	2.24 ±0.57	2.0 mg/kg *
Cd	0.14 ± 0.01	0.32 ± 0.04	0.16 ± 0.03	0.64 ±0.08	1.0 mg/kg **
Ni	0.41 ± 0.12	0.64 ± 0.15	0.19 ± 0.02	0.58 ± 0.10	-------------
Pb	0.09 ± 0.01	0.16 ± 0.03	0.19 ± 0.01	0.03 ± 0.03	1.5 mg/kg **

* obtained from [24], ** obtained from [35].

**Table 3 ijerph-18-10023-t003:** Total lipids (as g/100 g) and fatty acid composition of total lipids (as% of TFA) of wild and farmed *M. galloprovincialis* from the Black Sea (Bulgaria).

	VPS (Wild)	BSS (Farmed)	KS (Wild)	DFS (Farmed)
Total lipids (g/100g ww)	2.28 ± 0.11 ^a^	2.35 ± 0.23 ^a^	1.37 ± 0.06 ^b^	1.31 ± 0.22 ^b^
C12:0	tr	tr	tr	0.61 ± 0.03
C14:0	5.24 ± 0.17	3.17 ± 0.24	3.16 ± 0.29	1.07 ± 0.06
C15:0	0.63 ± 0.04	0.47 ± 0.04	tr	nd
C16:0	27.28 ± 0.82	18.78 ± 1.58	26.62 ± 0.51	16.28 ± 1.21
C17:0	0.89 ± 0.07	0.61 ± 0.03	tr	tr
C18:0	5.39 ± 0.26	4.44 ± 0.61	6.41 ± 0.84	4.19 ± 0.02
C20:0	tr	tr	0.17 ± 0.05	tr
C21:0	nd	tr	tr	tr
C22:0	nd	0.26 ± 0.04	tr	0.20 ± 0.01
C23:0	nd	tr	tr	tr
C24:0	0.49 ± 0.19	0.34 ± 0.03	0.44 ± 0.18	0.23 ± 0.02
SFA	40.05 ± 1.04 ^a^	28.19 ± 1.78	37.07 ± 1.37 ^a^	22.69 ± 1.21
C14:1	0.10 ± 0.01	tr	0.11 ± 0.04	0.10 ± 0.01
C16:1	30.07 ± 1.31	20.22 ± 1.10	8.89 ± 0.19	3.81 ± 0.02
C17:1	tr	0.17 ± 0.02	tr	tr
c-C18:1n-9	1.49 ± 0.12	1.25 ± 0.13	1.70 ± 0.16	2.57 ± 0.03
C20:1	nd	1.42 ± 0.13	0.81 ± 0.18	0.54 ± 0.02
C21:1	tr	tr	tr	0.35 ± 0.03
C24:1	nd	tr	0.21 ± 0.06	0.21 ± 0.05
MUFA	31.73 ± 1.35	23.21 ± 1.05	11.89 ± 0.05	7.60 ± 0.07
C18:2n-6	0.76 ± 0.06	0.51 ± 0.04	1.21 ± 0.03	2.06 ± 0.05
C18:3n-3	2.00 ± 0.69	0.92 ± 0.03	3.10 ± 0.14	1.10 ± 0.06
C20:2	0.75 ± 0.12	2.07 ± 0.11	0.55 ± 0.14	0.94 ± 0.03
C20:3n-6	nd	0.57 ± 0.10	2.27 ± 1.04	1.44 ± 0.06
C20:3n-3	tr	0.17 ± 0.01	0.27 ± 0.11	0.62 ± 0.11
C20:4n-6	1.44 ± 0.72	3.54 ± 0.88	5.70 ± 2.38	6.24 ± 0.19
C22:2	nd	0.23 ± 0.15	1.43 ± 0.46	0.82 ± 0.12
C20:5n-3	3.17 ± 0.10 ^a^	3.41 ± 0.27 ^a^	10.85 ± 1.14	31.70 ± 0.79
C22:6n-3	19.55 ± 2.21 ^b^	30.22 ± 2.08 ^a^	19.64 ± 2.24 ^b^	26.85 ± 0.88 ^a^
C22:5n-3	nd	5.41 ± 1.45	5.10 ± 0.94	nd
C22:4n-3	nd	1.54 ± 0.22	0.92 ± 0.36	nd
PUFA	27.75 ± 2.82	47.18 ± 3.47	51.04 ± 1.40	69.71 ± 1.20

Results represent mean values ± standard deviation (*n* = 3); SFA: saturated fatty acids; MUFA: monounsaturated fatty acids; PUFA: polyunsaturated fatty acids; nd—not detected; tr—trace levels were <0.1% of TFA; ^a,b^ values in a row not sharing a common superscript are significantly different (*p* < 0.05).

**Table 4 ijerph-18-10023-t004:** Fatty acid ratios and nutrition quality indices of total lipids of wild and farmed *M. galloprovincialis* from the Black Sea (Bulgaria).

	VPS (Wild)	BSS (Farmed)	KS (Wild)	DFS (Farmed)
PUFA/SFA	0.69 ± 0.07 ^c^	1.74 ± 0.21 ^b^	1.38 ± 0.09 ^b,c^	3.08 ± 0.21 ^a^
n-6/n-3	0.12 ± 0.03 ^a^	0.17 ± 0.08 ^a^	0.28 ± 0.10 ^a^	0.19 ± 0.01 ^a^
AI	0.82 ± 0.03 ^a^	0.44 ± 0.03 ^c^	0.63 ± 0.04 ^b^	0.27 ± 0.02 ^d^
TI	0.38 ± 0.03 ^a^	0.18 ± 0.01 ^c^	0.27 ± 0.00 ^b^	0.11 ± 0.01 ^d^
DHA + EPA (mg/100g EP)	428.8 ± 28.2 ^c^	655.6 ± 34.1 ^a^	309.4 ± 33.7 ^d^	556.5 ± 15.6 ^b^

Results represent mean values ± standard deviation (*n* = 3); ^a,b,c,d^ values in a row not sharing a common superscript are significantly different (*p* < 0.05); EP—edible portion; AI—atherogenicity index; thrombogenicity index—TI; AI and TI are calculated using the formulas described in [11].

**Table 5 ijerph-18-10023-t005:** Estimated target hazard quotient (THQ) and target cancer risk (TR) values of wild and farmed *M. galloprovincialis* from the Black Sea (Bulgaria).

Element	RfD * (mg/kg/d)	Estimated Target Hazard Quotient (THQ)				CSFo ** (mg/kg/d)	Estimated Target Risk (TR)			
		VPS (Wild)	BSS (Farmed)	KS (Wild)	DFS (Farmed)		VPS (Wild)	BSS (Farmed)	KS (Wild)	DFS (Farmed)
As	0.0003	0.005	0.057	0.023	0.082	1.75	6.0 × 10^−6^	13 × 10^−6^	5.1 × 10^−6^	18 × 10^−6^
Cd	0.001	0.002	0.004	0.002	0.007	NA	NA	NA	NA	NA
Cr	0.003	0.001	0.001	0.0004	0.001	0.5	3.6 × 10^−7^	9.0 × 10^−7^	2.4 × 10^−7^	8.2 × 10^−7^
Cu	0.040	0.001	0.001	0.001	0.001	NA	NA	NA	NA	NA
Fe	0.700	0.002	0.003	0.002	0.003	NA	NA	NA	NA	NA
Ni	0.020	0.0002	0.0004	0.0001	0.0003	1.7	3.2 × 10^−6^	5.1 × 10^−6^	1.5 × 10^−6^	4.6 × 10^−6^
Pb	0.0035	0.0003	0.0005	0.0006	0.0010	0.0085	3.8 × 10^−9^	6.7 × 10^−9^	8.2 × 10^−9^	1.4 × 10^−8^
Zn	0.300	0.001	0.001	0.001	0.001	NA	NA	NA	NA	NA
HI	----	0.006	0.010	0.006	0.014					

* Oral reference doses (RfD) were obtained from the EPA Region III Risk-Based Concentrations summary table [25], with the exception of Pb [72]; The body weight used for calculation is 70 kg; HI-Hazard Index; ** CSFo, the oral carcinogenic slope factor obtained from the integrated risk information system database [25]; NA, not available.

**Table 6 ijerph-18-10023-t006:** Hazard quotients, HQ_EFA_, for benefit–risk ratio of EFA vs. toxic/essential elements and arsenic for intake of wild and farmed *M. galloprovincialis* from the Black Sea (Bulgaria).

*M. galloprovincialis*	As	Cd	Cr	Cu	Fe	Ni	Pb	Zn
VPS (wild)	0.11	0.21	0.07	0.10	0.25	0.03	0.04	0.09
BSS (farmed)	0.17	0.35	0.14	0.06	0.25	0.03	0.05	0.07
KS (wild)	0.14	0.37	0.08	0.16	0.40	0.02	0.13	0.16
DFS (farmed)	0.29	0.81	0.11	0.06	0.34	0.04	0.12	0.13

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
