# Peer review of "Trace Elements and Omega-3 Fatty Acids of Wild and Farmed Mussels (Mytilus galloprovincialis) Consumed in Bulgaria: Human Health Risks"

_ijerph, 2021, doi:10.3390/ijerph181910023_

Round 1

Reviewer 1 Report

Trace elements and omega-3 fatty acids are quite important for human health, This paper studies the toxic and essential element (As, Cd, Cr, Cu, Fe, Mn, Ni, Pb and Zn) concentration and fatty acid composition of mussel consumed in Bulgaria. The paper has good logical organization and nice writing style, the approach of this paper is interesting, but, there are some flaws in the methodology that need dealing with. So, I would like to suggest the authors to make a major revision and reconsideration on this article. Some questions are listed below:

  1. Line 86: the enrichment of essential elements is related to time, the age of the samples (mussel) must be clearly stated, especially wild samples.
  2. Line 22, 125: The acronym ( EDI, THQ, HI) should be explained the first time that appears.
  3. As the author said: Major fatty acid levels in marine organisms generally depend on numerous biotic and abiotic factors such as water temperature, salinity, UV radiation, food availability, etc. But, what is “natural environment”(Line 89) ? The growth environment of the samples needs more detailed description .
  4. Line 102: “Each tissue sample (around 1 g wet weight) was weighed......”, Whether there is a content difference in the part of the tissue, if there is, the sampling location should be stated. In the meantime, sampling method should be detailed.
  5. Line 265: Table 3. Total lipids  and fatty acid composition of total lipids of wild and farmed mussel, C20:5n-3and C22:6n-3 of farmed samples were significantly higher than the wild ones, Comparison is made between the farmed and the wild, but there is no discussion of the large differences.

Author Response

Trace elements and omega-3 fatty acids are quite important for human health, This paper studies the toxic and essential element (As, Cd, Cr, Cu, Fe, Mn, Ni, Pb and Zn) concentration and fatty acid composition of mussel consumed in Bulgaria. The paper has good logical organization and nice writing style, the approach of this paper is interesting, but, there are some flaws in the methodology that need dealing with. So, I would like to suggest the authors to make a major revision and reconsideration on this article. Some questions are listed below:

We appreciate the kind statements of the reviewer. The responses to the comments are as follows and changes are highlighted with Grey colour in the manuscript.

  1. Line 86: the enrichment of essential elements is related to time, the age of the samples (mussel) must be clearly stated, especially wild samples.

# Since the purpose of the research is to evaluate the human health risk of the wild and farmed mussel consumption, only mussels of commercial size were subjected to analysis (between 42.24 and 51.06 mm), thus avoiding differences raleted to age and size.

  1. Line 22, 125: The acronym ( EDI, THQ, HI) should be explained the first time that appears.

# It has been corrected in edited manuscript

  1. As the author said: Major fatty acid levels in marine organisms generally depend on numerous biotic and abiotic factors such as water temperature, salinity, UV radiation, food availability, etc. But, what is “natural environment”(Line 89) ? The growth environment of the samples needs more detailed description.

# Samples were obtained from two of the largest mussel farms along the Bulgarian Black Sea coast and from local fishermen (for the wild specimens). Mussels were of commercial size or “market-ready”. External factors (temperature, salinity, etc.) were not considered, because this was not an accent in our study. However, if editor or reviewers recommend that, it could be done, based on literature information.

  1. Line 102: “Each tissue sample (around 1 g wet weight) was weighed......”, Whether there is a content difference in the part of the tissue, if there is, the sampling location should be stated. In the meantime, sampling method should be detailed.

# The mussel samples were rapidly washed and the meat (muscle tissue) was removed from the shell with Teflon knife in order to avoid any contamination. Each tissue sample (around 1 g wet weight) was weighed, placed in Teflon digestion vessels for further acid wet digestion. Since the purpose of the study is to investigate the human health risk, the edible portion of the farmed and wild mussels were analysed (e.i. the one that consumer eat). The farmed mussels were collected from two mussel farms while the wild ones - from their substrates (rocks) by local fishermen on the places donated on the map in the maniscript (Figure 1).

  1. Line 265: Table 3. Total lipids  and fatty acid composition of total lipids of wild and farmed mussel, C20:5n-3and C22:6n-3 of farmed samples were significantly higher than the wild ones, Comparison is made between the farmed and the wild, but there is no discussion of the large differences.

# Mussels cannot synthesize LC-PUFAs, such as DHA, de novo and therefore they are obtained from the diet. Possible reasons for the differences in n-3 LC-PUFA contents are discussed (lines 329-332)

Reviewer 2 Report

In this paper, the author try to find the difference and human health risk of trace elements and omega-3 fatty acids of wild and farmed mussels.  I think the methd and final results are fine and show some interesting idea for readers in this version. So, I do have no further comment. Just take care of the grammar mistake and check them in detail.

Author Response

In this paper, the author try to find the difference and human health risk of trace elements and omega-3 fatty acids of wild and farmed mussels.  I think the methd and final results are fine and show some interesting idea for readers in this version. So, I do have no further comment. Just take care of the grammar mistake and check them in detail

We appreciate the kind statement of the reviewer. Grammar was checked in detail.

Reviewer 3 Report

Overall this is an extremely well-written manuscript and should be published nearly as is.  Below are a few comments for the authors to consider.

line                                                                         Comments

19                           Consider beginning The Abstract with:  The Black ………

22                    What are the abbreviations  “EDI, THQ, HI, TR” ?  Don’t seem to be                                             analytical methods like “AA, ICP, ICPMS..” .  Consider spelling out these                                     abbreviations and put a comma after the last.

27                    What are these abbreviations,  PUFA/SFA and SFA, AI and TI?      

35                    Why are n-3 LCPUFAs important? For example, neurological development,                                  improved cognition, cardiovascular disease (CVD) benefits.

65                    Which elements are essential and which toxic?    Or are all al toxic above a                                    certain concentration?  They are listed in Tables 1 and 2.

66                    concentrations

68                    like on line 22 what are these abbreviations?        THQ, HI and TR

215                  An important reference not cited by the authors is “The mussel watch                                             intercomparison of trace level constituent determinations” by  W. B. Galloway                              and others in 1983.

Author Response

Overall this is an extremely well-written manuscript and should be published nearly as is.  Below are a few comments for the authors to consider.

We appreciate the kind statements of the reviewer. The responses to the comments are as follows and changes are highlighted with Yellow colour in the manuscript.

Line                  Comments

19                    Consider beginning The Abstract with:  The Black ………                     

# It has been corrected in edited manuscript

22                    What are the abbreviations “EDI, THQ, HI, TR”?  Don’t seem to be analytical methods like “AA, ICP, ICPMS”.  Consider spelling out these abbreviations and put a comma after the last.

# The abbreviations are fully written at the abstract paragrpham. There are detailly explained in the Material and Methods section

27                    What are these abbreviations, PUFA/SFA and SFA, AI and TI?             

# the abbreviations are introduced in the main body text of the manuscript. If the reviewer insist they will be included in the Abstract Section too

35                    Why are n-3 LCPUFAs important? For example, neurological development, improved cognition, cardiovascular disease (CVD) benefits.                                               

# The importance of n-3 LC-PUFAs was highlighted in the introduction

65                    Which elements are essential and which toxic? Or are all al toxic above a certain concentration? They are listed in Tables 1 and 2.

         # A new detailed information is included in the text reffering Line 66-69

66                    concentrations                                                                                   

#  It has been corrected in edited manuscript

 68                    like on line 22 what are these abbreviations?  THQ, HI and TR

#  It has been corrected in edited manuscript

215                  An important reference not cited by the authors is “The mussel watch intercomparison of trace level constituent determinations” by  W. B. Galloway and others in 1983.

# All the references were checked and the missing references were added at the appropriate place. Some references were removed and replaced with better sources.

Reviewer 4 Report

The manuscript is well conducted and is covering an interesting issue. However, there are some minor revision that I recommend to consider. 

In the introduction section, is recommended to explain little about the toxic effects of the analyzed elements and if any of this have cause human damage from the consumption of this sea foods. 

In the material and methods, the map is plain. Could you improve another map more "visual"? 

Some reagents are not properly indicated (brand, country, purity...) 

Conclusion: what does the authors think about the future concerns ? Is adequate to eat this sea food? 

Author Response

The manuscript is well conducted and is covering an interesting issue. However, there are some minor revision that I recommend to consider. 

We appreciate the kind statements of the reviewer. The responses to the comments are as follows and changes are highlighted with Blue colour in the manuscript.

  1. In the introduction section, is recommended to explain little about the toxic effects of the analyzed elements and if any of this have cause human damage from the consumption of this sea foods. 

# An updated detailed information about the toxic and essential elements is included in the edited manuscript reffering Line 66-69 (Introduction part)

  1. In the material and methods, the map is plain. Could you improve another map more "visual"? 

# The map is a subject to copywrite license. Using any other map (s) will be unpropriate for scientific purpose.

  1. Some reagents are not properly indicated (brand, country, purity...) 

# It has been corrected in edited manuscript

  1. Conclusion: what does the authors think about the future concerns? Is adequate to eat this sea food? 

# “Future studies on M. galloprovincialis trace elements, EPA and DHA contents need to be periodically carried out in order to provide up-to-date information for benefit-risk assessment of the only representative of the Bulgarian marine aquaculture”. 

“According to the estimated hazardous quotients and indexes, the levels of the analyzed wild and farmed mussles collected in the studied region can be considered relatively safe and mussels are suitable for consumption.”

This statement will be included in the Conclusion Section of the edited manuscript

Round 2

Reviewer 1 Report

The interpretation of the related issues can be understood.  Samples acquisition and processing, but also describe more accurate.